# Impact of Lung Compliance on Neurological Outcome in Patients with Acute Respiratory Distress Syndrome Following Out-of-Hospital Cardiac Arrest

**DOI:** 10.3390/jcm9020527

**Published:** 2020-02-14

**Authors:** June-sung Kim, Youn-Jung Kim, Muyeol Kim, Seung Mok Ryoo, Chang Hwan Sohn, Shin Ahn, Won Young Kim

**Affiliations:** Department of Emergency Medicine, University of Ulsan College of Medicine, Asan Medical Center, Seoul 05505, Korea; jsmeet09@gmail.com (J.-s.K.); yjkim.em@gmail.com (Y.-J.K.); kurt0217@naver.com (M.K.); chrisryoo@naver.com (S.M.R.); schwan97@gmail.com (C.H.S.); ans1023@gmail.com (S.A.)

**Keywords:** cardiac arrest, acute respiratory distress syndrome, mechanical ventilation, ICU management

## Abstract

(1) Background: Acute respiratory distress syndrome (ARDS) following cardiac arrest is common and associated with in-hospital mortality. We aim to investigate whether lung compliance during targeted temperature management is associated with neurological outcome in patients with ARDS after out-of-hospital cardiac arrest (OHCA). (2) Methods: This observational study is conducted in the emergency intensive care unit from January 2011 to April 2019 using data from a prospective patient registry. Adult patients (age ≥18 years) who survived non-traumatic OHCA and subsequently developed ARDS based on the Berlin definition are included. Mechanical ventilator parameters such as plateau pressure, tidal volume, minute ventilation, positive end expiratory pressure, and compliance are recorded for 7 days or until death, and categorized as maximum, median, and minimum. The primary outcome is a favorable neurological outcome defined as a Cerebral Performance Category score of 1 or 2 at hospital discharge. (3) Results: Regarding 246 OHCA survivors, 119 (48.4%) patients developed ARDS. A favorable neurologic outcome was observed in 23 (19.3%). Patients with a favorable outcome have a significantly higher lung compliance (38.6 mL/cm H_2_O versus 27.5 mL/cm H_2_O), lower inspiratory pressure (12.0 cm H_2_O versus 16.0 cm H_2_O), and lower plateau pressure (17.0 cm H_2_O versus 21.0 cm H_2_O) than those with a poor neurologic outcome (all *p* < 0.01). Concerning time-dependent cox regression models, all maximum (adjusted hazard ratio [HR] 1.05, 95% confidence interval [CI] 1.02–1.09), minimum (HR 1.08, 95% CI 1.03–1.13), and median (HR 1.06, 95% CI 1.02–1.09) compliances are independently associated with a good neurologic outcome. Maximum compliance, >32.5 mL/cm H_2_O at day 1, has the highest area under the receiver operating characteristic curve (0.745) with a positive predictive value of 90.4%. (4) Conclusions: Lung compliance may be an early predictor of intact neurologic survival in patients with ARDS following cardiac arrest.

## 1. Introduction

Despite recent advances in prevention and resuscitation, the mortality and morbidity of cardiac arrest (CA) remain high [1,2,3]. Postcardiac arrest syndrome (PCAS), the major cause of death in CA patients, is characterized by a systemic post-arrest ischemia-reperfusion injury with activation of the inflammatory response; its in-hospital mortality is >50% [4]. PCAS can affect multiple organs. Pulmonary involvement manifests as acute respiratory distress syndrome (ARDS), which often results in poor clinical outcomes [5]. The authors postulate that ARDS accelerates ischemic brain injury caused by aggravating the mismatch of supply and demand of oxygen (O_2_) and carbon dioxide (CO_2_) [5].

Mechanical ventilation is the cornerstone of managing and treating ARDS. However, it can also cause parenchymal lung damage through over-distension and cyclic alveolar opening and closing [6,7,8]. Lung-protective ventilation utilizing a lower tidal volume (V_T_) and higher positive end-expiratory pressure (PEEP) has shown to improve survival in numerous trials [9]. Even though the benefit of interventions has been controversial, guidelines have suggested that fluid restriction, use of neuromuscular blocking agents, and prone ventilation could reduce mortality in severe ARDS patients [10]. Moreover, a recent study introduced that a lung could modulate the neurological injury and recovering from ARDS could reduce brain damage [11]. Therefore, discovering accurate prognostic markers of ARDS severity is essential to facilitate risk stratification and apply novel therapeutic interventions to improve outcomes and predict neurologic outcomes among CA survivors. Since lung compliance correlates with the aerated functional component of the pulmonary system [12], we hypothesize that lung compliance, an easily measured clinical variable, could be a more important prognostic marker than V_T_ or PEEP in patients with ARDS after CA.

This study aims to investigate whether lung compliance during targeted temperature management (TTM) is associated with the neurological outcome in patients with ARDS after out-of-hospital cardiac arrest (OHCA).

## 2. Materials and Methods

### 2.1. Study Design and Population

This retrospective study used prospectively collected data that were entered into the Asan Medical Center emergency intensive care unit out-of-hospital cardiac arrest (OHCA) patient registry from January 2011 to April 2019. Asan Medical Center is a tertiary referral academic center in Seoul, Korea, with 2700 beds and approximately 130,000 Emergency Department patient visitations annually. The study was approved by the institutional research ethics committee; the requirement for informed consent was waived because the study was retrospective in nature (study number 2019-1187).

The OHCA registry consists of all adults (age ≥18 years) admitted to the emergency intensive care unit with non-traumatic cardiac arrest (CA). Patients with the following criteria are excluded from the registry: age <18 years, presumed traumatic CA, “do not resuscitate” status, transfer to another hospital, and declined proper treatment. This study enrolled patients from the registry who developed acute respiratory distress syndrome (ARDS) within 48 h after CA. We identified ARDS patients using the Berlin definition. Briefly, we found those who had classic risk factors for the development of ARDS by reviewing medical charts, had two consecutive arterial blood gas analysis (ABGA) with partial pressure of arterial oxygen (PaO_2_)/fraction of inspired oxygen (FiO_2_) (P/F) ratio <300 mmHg in a mechanically ventilated patient, had bilateral radiographic opacities on chest radiography or on computed tomography that could not be explained by effusions, pulmonary collapse, or nodules, and had respiratory failure that could not be explained by cardiogenic pulmonary edema. All chest images were reviewed by radiologists. The severity of the ARDS was determined based on the degree of hypoxemia: mild (200 mmHg < PF ratio ≤300 mmHg), moderate (100 mmHg < PF ratio ≤200 mmHg), and severe (PF ratio <100 mmHg). We excluded patients who received extracorporeal membrane oxygenation (ECMO) during or after cardiopulmonary resuscitation (CPR). Moreover, we excluded the cases of cardiogenic pulmonary edema, which were defined as left ventricular ejection fractions below 30% based on transthoracic echocardiography.

### 2.2. Routine Post-Cardiac Arrest Care Protocol

Regarding all patients, CPR and post-resuscitation care, including coronary intervention and target temperature management (TTM), were administered following the current Advanced Cardiac Life Support guidelines. TTM consists of three phases: induction, maintenance, and rewarming. TTM was induced with a cooling device with self-adhesive hydrogel coated pads (Arctic Sun Energy Transfer Pads, Medivance Corp, Louisville, CO, USA). The target temperature of 33 °C was maintained for 24 h (maintenance phase), and patients were then rewarmed to 36.5 °C at a rate of 0.25 °C/h. The temperature was recorded before starting TTM and monitored using an esophageal probe, and continuous intravenous propofol and opioids (morphine or remifentanil) were administered for sedation and analgesia during TTM. Neuromuscular blocking agents were used upon the return of spontaneous circulation and achieved the target temperature to reduce shivering and keep synchronizing respiration of patients with the ventilator. Patients were deeply sedated (defined by a score on the Richmond Agitation-Sedation Scale of −2 to −4) to prevent shivering and to avoid awareness during TTM. After finishing the rewarming phase, light sedation (defined by a score on the Richmond Agitation-Sedation Scale of 0 or −1) was warranted. Arterial blood gas analysis (ABGA) was performed in all patients just after the return of spontaneous circulation (ROSC) and repeated every 2 ± 1 h. The initial lactate level just after ROSC was performed for every survivor. Additionally, all patients were mechanically ventilated in pressure-controlled mode and every change of ventilator parameter, such as tidal volume (V_T_), peak end expiratory pressure (PEEP), inspiratory pressure, plateau pressure (P_plat_), respiratory rate, and minute ventilation was recorded in the electronic medical record by experienced intensive care unit nurses.

### 2.3. Definition of Variables

As part of the protocol, we performed a chest radiograph at least once- a day and obtained ABGA every hour during induction and every 2 h during the maintenance period. Using the results of ABGA and ventilator parameters, the P/F ratio and compliance (V_T_/[P_plat_–PEEP]) were calculated. Additionally, we categorized the calculated measures, including V_T_, minute volume, and compliance as maximum, median, and minimum for 7 days or until death, whichever came first. When there were missing ABGA or ventilator variable data (i.e., performed ABGA less than 3 times because patients visited the Emergency Department (ED) just before the end of the day), the means of each variable were calculated by max and minimum values of each patient and filled before the statistical analysis (N = 9).

We extracted demographic and baseline clinical data from the registry, including age, sex, and underlying diseases. Additionally, we gathered other CA data according to the Utstein Style recommendations, including location, witnessed arrest, bystander CPR, the time gap between the perception of cardiac arrest and Advanced Cardiac Life Support, duration of CPR, initial documented rhythm, presumed arrest cause, and initial electrocardiography rhythm. Based on the results of the chest images, laboratory data, and needs of vasopressors, the traditional risk factors of developing ARDS, such as shock, aspiration, pancreatitis, pneumonia, pulmonary contusion, and sepsis were recorded by physicians on duty. The Sequential Organ Failure Assessment (SOFA) score was determined from the initial data on admission. The primary outcome of the study was the association of lung compliance with the favorable neurologic outcome at discharge (defined as Cerebral Performance Category (CPC) score of 1 or 2). The CPC score ranged from 1 to 5, with 1 presenting normal brain function and 5 presenting brain death. This score was routinely assessed and recorded in electronic medical records by well-trained physicians on duty in the emergency intensive care unit. Other ventilator parameters were compared between patients with the good and poor neurologic outcomes as secondary outcomes.

### 2.4. Statistical Analyses

Statistical analyses were performed using SPSS Statistics for Windows version 23 (SPSS Inc., Chicago, IL, USA). Continuous variables were expressed as medians with interquartile ranges. Categorical variables were analyzed with the chi-square or Fisher’s exact tests. The normality of distribution was examined using the Kolmogorov-Smirnov test. The Mann–Whitney U test was used for comparisons between the good and poor neurologic outcome groups. To examine the association between ARDS and poor neurologic outcome, we used a Cox regression model with maximum, median, and minimum lung compliances as a time-varying covariate. Additionally, sensitivity, specificity, positive predictive value, negative predictive value, positive likelihood ratio, and negative likelihood ratio were calculated by standard statistical methods. *p* <0.05 was considered significant.

## 3. Results

During the study period, a total of 295 registry patients survived cardiac arrest (CA), and 246 were analyzed after excluding 49 who received extracorporeal membrane oxygenation (ECMO) and had cardiogenic pulmonary edema (Figure 1). Among these, 119 patients (48.4%) developed acute respiratory distress syndrome (ARDS) within 48 h (20 mild, 48 moderate, 51 severe). Regarding the 119 patients with ARDS, 23 (19.3%) had a favorable neurologic outcome at discharge. Concerning 96 patients with poor outcomes, 59 ARDS patients were in-hospital deaths. Considering these, 37 patients (62.7%) died due to ongoing multiorgan failure and refractory shock within 72 h (i.e., they could not finish target temperature management [TTM]). Six patients (10.2%) had both severe a hypoxic brain injury and progression of multiorgan failure. Additionally, 16 patients (27.1%) stopped treatment due to severe hypoxic brain injury and death.

### 3.1. Baseline Characteristics

The baseline characteristics of the study population are presented in Table 1. The median age was 67 years, with male predominance (63.9%). Body mass index and the proportion of smokers were not different between the two groups. Hypertension (48.7%) was the most frequent comorbidity, followed by diabetes (37.0%). The favorable neurologic outcome group had a more shockable initial rhythm (64.3% versus 16.0%; *p* <0.001), more presumed cardiac cause (65.2% versus 37.5%; *p* = 0.016), and shorter CPR duration (11.0 versus 31.5 min; *p* <0.001). Other comorbidities and characteristics of CA did not differ significantly between the two groups. Among classic risk factors of developing ARDS, pulmonary contusion (71.4%) was the most frequent, followed by shock (58.0%) and aspiration (52.1%). Shock was more often in the poor outcome group (64.6% versus 30.4%, *p* = 0.003). Both groups showed similar body temperature before starting TTM, and all patients had a TTM of 33 °C during maintenance. Besides propofol, all of the sedative and neuromuscular blocking agents were used similarly between both groups. Initial lactate levels after return of spontaneous circulation (ROSC) were significantly higher in patients with the poor neurologic outcome (13.2 mmol/L versus 9.3 mmol/L, *p* = 0.002). Furthermore, patients with ARDS showed a higher proportion of poor neurologic outcome (81.5% versus. 60.6%, *p* <0.001) and higher all causes mortality (61.0% versus. 37.8%, *p* <0.001).

### 3.2. Mechanical Ventilator Parameters on Admission and Neurologic Outcome

The result of ABGA and mechanical ventilator settings on admission for each patient group are summarized in Table 2. The good neurologic outcome group showed a higher FiO_2_ level (100.0% versus 100%, *p* = 0.028), lower PaCO_2_ level (46.0 mmHg versus 57.0 mmHg; *p* = 0.005) and higher pH (7.2 versus. 7.0, *p* = 0.001) than that of the poor outcome group. Regarding ventilator settings, patients with a good outcome showed significantly lower requirements of inspiratory pressure (12 cm H_2_O versus. 16 cm H_2_O; *p* < 0.001), P_plat_ (17.0 cm H_2_O versus. 21.0 cm H_2_O; *p* < 0.001), and respiratory rate (18.0/min versus. 20.0/min; *p* = 0.013). Moreover, compliance was significantly higher in the favorable outcome group (38.6 mL/cm H_2_O versus. 27.5 mL/cm H_2_O; *p* < 0.001).

### 3.3. Compliance and Good Neurologic Outcome

Table 3 shows the time-dependent regression model for predicting the neurologic outcome. All of the maximum, median, and minimum lung compliances were independent risk factors for the favorable neurologic outcome at discharge. Among these variables, the minimum value was most significantly associated with the neurologic outcome (adjusted hazard ratios (HR) 1.08, 95% confidence interval (CI) 1.03–1.13; *p* <0.001).

### 3.4. Differences of Day-1 Compliance for Neurologic Outcome

Overall trends of maximum, minimum, and median compliances by the neurologic outcome group over time are shown in the additional file [see Additional file, Appendix A]. Regarding all seven compliances, day 1 showed significant large differences between the good and poor outcome groups which decreased over 7 days. Figure 2 shows the differences of ROSC, day-1 maximum, median, and minimum compliances between the good and poor outcome groups. Considering sensitivity and specificity analysis (Table 4), maximum compliance above 32.5 on day 1 presented the highest specificity (72.0%), positive likelihood ratio (2.48), and positive predictive value (90.4%) compared to that of other compliances.

## 4. Discussion

During this registry-based study, we found that lung compliance was associated with intact neurologic survival using time-dependent regression analysis. Maximum, minimum, and median compliance showed similar diagnostic performance for predicting good neurologic outcomes and had the highest predictive value on day 1. Even though causality can be inferred only from randomized controlled trials, our finding suggested that the degree of the mechanical property of the respiratory system was an important marker for risk stratification in post-cardiac arrest syndrome (PCAS) management.

Our neurologic intact survival rate of acute respiratory distress syndrome (ARDS) patients was lower than previous studies, which reported rates ranging from 29.8–53.2% [13,14,15]. The incidence of ARDS following cardiac arrest (CA) has not been well reported; it varies from 5–65% depending on how ARDS is defined [8,16]. However, it may be common due to the overlap between the pathophysiology of ARDS and PCAS. Lung contusion, ischemia, and exposure to high-dose oxygen during CPR and the following reperfusion after CA, a profound systemic inflammatory response, ventilator-associated injuries, secondary infections, and systemic immune reactions could contribute to the development of ARDS [17,18]. These risk factors also existed in our study population and supported the prior findings that ARDS was a common problem for survivors of cardiac arrest [see Additional file, Appendix A]. Even though the clinical impact of ARDS on PCAS patients is not fully known, a growing body of evidence has been cumulated showing that the occurrence of ARDS after CA has a negative impact on both survival and recovery of neurologic impairments [19,20,21]. Thus, early recognition of ARDS after CPR may allow for prompt initiation of treatment known to improve outcomes, such as low tidal volume ventilation, prone positioning, and continuous neuromuscular blockades [8,22,23,24].

We identified the correlations between higher lung compliances and favorable neurologic outcomes. These results could be explained by multifactorial reasons. First, lung compliance could reflect the severity of the ARDS. Numerous reports announced that patients with severe ARDS itself had a higher mortality in critically ill patients, not only in patients with PCAS [12]. Moreover, in the landmark ARDS Network trial, limiting low plateau pressure (P_plat_) resulted in higher lung compliance and improved outcome [25]. Second, severe ARDS aggravated the unbalance levels of oxygen and carbon dioxide, which could impair recovering neurocognitive function. However, the exact mechanisms of O_2_ and CO_2_ are not fully known. Although prior study showed post-resuscitation hyperoxia in the first 24 h was associated with a worse outcome [26], a recent randomized trial revealed that different target levels of PaCO_2_ (low-normal [33–35 mmHg] versus. high-normal [43–45 mmHg]) or PaO_2_ (normoxia [75–112 mmHg] versus. moderate hyperoxia [150–190 mmHg]) did not change the concentration of neuron-specific enolase at 48 h after ROSC [27]. These mismatches might largely be due to different inclusion in the study population or unmeasured confounders such as different prehospital settings [26,27]. Concerning our result, the favorable outcome group also showed a shorter duration of CPR and a higher proportion of shockable rhythm, which was common in the population of the favorable neurologic outcome [3]. However, after modifying these effects using regression analysis, we found that the higher compliances were associated with good outcome. Last, patients with lower lung compliance could be less promising due to ventilator-induced lung injuries like barotrauma [28].

Additionally, we found that lung compliance, whether maximum, minimum, or median, was associated with the neurologic outcome at discharge. We supposed that lung compliances were changing dynamically during intensive care. Each of max, minimum, and median compliance could reflect the best, worst, mean condition of lung elasticity. We tried to evaluate which conditions were associated with clinical outcomes. Although there were slight differences in the hazard ratio among three compliances, the degree was not clinically discriminated. Regarding the timing of measuring, the compliance difference between groups was dramatic in the first 24 h after admission and diminished over time. Despite the casualty not being confirmed, these results imply that the recovery of brain injury was determined in the initial phase, and proper management of ARDS would focus on the first 24 h. This is consistent with evidence that brain damage started accelerating after 24 h from global ischemia in laboratory animal and human tests [29,30]. When we set the cut-off value for each compliance, tests revealed a high positive predictive value with low negative predictive value. Although numerous studies have focused on discovering factors for poor neurologic outcomes, it is essential to develop strategies for predicting good neurologic outcomes among out-of-hospital cardiac arrest survivors to appropriately tailor medical therapies for each patient [13,14].

This study had several limitations. First, due to its retrospective design, our results may not be generalized to other circumstances. Moreover, definite casualties between compliance and the neurologic outcome could not be confirmed. Second, because the diagnosis of ARDS was based on the P/F ratio, chest images, and echocardiography, ARDS incidence might be overestimated because hydrostatic pulmonary edema could not be found based on these examinations. However, the Berlin definition allowed some flexibility, including combining cardiogenic pulmonary edema because it is impossible to exclude all pure cardiogenic causes, even when measuring cardiac function through echocardiography. Moreover, some post-cardiac arrest patients can have concomitant ARDS and cardiogenic pulmonary edema. To strengthen our result, we additionally performed a sensitivity analysis without excluding cardiac failure. Although specific values of hazard ratio changed, we found a similar trend in the results that compliance was an independent risk factor for predicting neurologic outcomes [see Additional file. Appendix A]. Third, we excluded patients who received extracorporeal membrane oxygenation, had a terminal illness or declined proper management, which may have introduced selection bias. Fourth, we did not consider other confounding factors, such as the quality of initial CPR, combined infections, and the use of vasopressors that may have influenced the results. Fifth, due to the relatively small sample size, we could collect only 7-days follow-up data. Measuring compliance is impossible when patients extubated or died. When we followed-up our study population, there were 119 ARDS patients on day-1, but only 30 patients remained with mechanical ventilation on day-7. Future studies with a bigger sample size may reveal long-term clinical impacts of ARDS on patients with PCAS. Sixth, nine patients performed only two ABGAs on hospital day-1 because they were admitted to the Emergency Department just before the end of the day. We calculated compliances and divided them into max and minimum without a median. Since there was a relatively small sample size, we included nine patients with filling mean values for missing median values. To make sure that these missing variables would not influence the results, we additionally performed a sensitivity analysis abstracting these nine patients and confirmed that the trends of the results were not changed. Finally, the sedative and neuromuscular blocking agent dosages, duration, and abdominal pressure for each patient were not controlled, which could have had an impact on calculated compliance.

## 5. Conclusions

Accompanying shockable rhythm and short duration of CPR, lung compliance may be an early independent predictor of intact neurologic survival in acute respiratory distress syndrome patients following cardiac arrest.

## Figures and Tables

**Figure 1 jcm-09-00527-f001:**
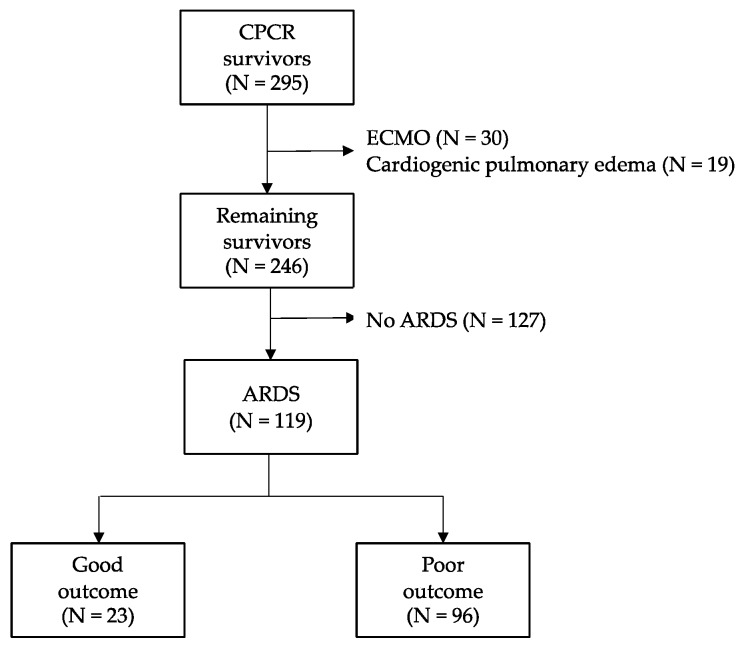
Flow chart of the study population. Abbreviations: CPCR = cardiopulmonary cerebral resuscitation; ECMO = extracorporeal membrane oxygenation; ARDS = acute respiratory distress syndrome.

**Figure 2 jcm-09-00527-f002:**
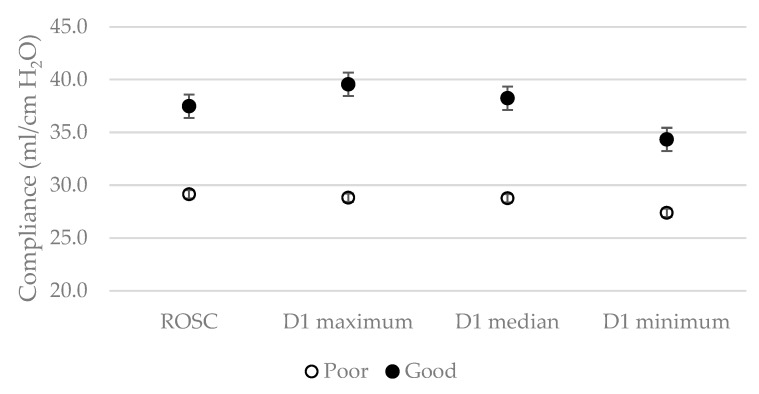
Differences of compliances on day-1 depend on neurologic outcomes. Abbreviations: ROSC = return of spontaneous circulation; D = day.

**Table 1 jcm-09-00527-t001:** Baseline characteristics of the ARDS patients.

Characteristics	Total (n = 119)	Poor Outcome (n = 96)	Good Outcome (n = 23)	*p*-Value
Age	67.0 (57.0–78.0)	69.0 (61.0–77.0)	62.0 (52.0–80.5)	0.770
Male	76 (63.9)	59 (61.5)	17 (73.9)	0.264
BMI (kg/m^2^)	23.7 (21.1–26.1)	22.9 (20.8–26.0)	23.3 (18.9–26.2)	0.719
Smoker	44 (37.0)	33 (34.4)	11 (47.8)	0.235
Past history				
Previous ACS	21 (17.6)	16 (16.7)	5 (21.7)	0.567
Previous PCI	15 (12.6)	11 (11.5)	4 (17.4)	0.441
Hypertension	58 (48.7)	50 (52.1)	8 (34.8)	0.136
Diabetes mellitus	44 (37.0)	38 (39.6)	6 (26.1)	0.228
COPD	12 (10.1%)	12 (12.5)	0 (0.0)	0.075
CKD	23 (19.3)	20 (20.8)	3 (13.0)	0.560
Malignancy	14 (11.8)	12 (12.5)	2 (8.7)	0.599
Characteristics of Cardiac arrest				
Witnessed arrest	83 (69.7)	65 (67.7)	18 (78.3)	0.324
Bystander CPR	78 (65.5)	63 (65.6)	15 (65.2)	0.814
Shockable rhythm	22 (23.2)	13 (16.0)	9 (64.3)	<0.001
Cardiac caused	51 (42.9)	36 (37.5)	15 (65.2)	0.016
Time of CA to resuscitation (min)	5.0 (0.0–9.0)	5.0 (0.0–9.0)	4.0 (0.0–6.5)	0.708
Total CPR time (min)	29.0 (15.0–41.0)	31.5 (21.3–43.8)	11.0 (4.0–21.0)	<0.001
ARDS risk factors				
Shock	69 (58.0)	62 (64.6)	7 (30.4)	0.003
Aspiration	62 (52.1)	51 (53.1)	11 (47.8)	0.648
Pancreatitis	3 (2.5)	3 (3.1)	0 (0.0)	0.253
Pneumonia	44 (37.0)	36 (37.5)	8 (34.8)	0.808
Pulmonary contusion	85 (71.4)	69 (71.9)	16 (69.6)	0.827
Sepsis, no pneumonia	19 (16.0)	16 (16.7)	3 (13.0)	0.664
Sedative and NMB during TTM				
Midazolam	26 (21.8)	22 (22.9)	4 (17.4)	0.565
Ketamine	2 (1.7)	2 (2.1)	0 (0.0)	0.352
Morphine	38 (31.9)	31 (32.3)	7 (30.4)	0.863
Propofol	80 (67.2)	59 (61.5)	21 (91.3)	0.003
Fentanyl	58 (48.7)	45 (46.9)	13 (56.5)	0.405
Continuous NMB	54 (45.4)	42 (43.8)	12 (52.2)	0.467
ICU care factors				
BT, before TTM applied (°C)	35.9 (35.1–36.5)	35.8 (35.0–36.5)	35.8 (35.2–36.2)	0.521
Lactate, after ROSC (mmol/L)	12.7 (9.4–15.0)	13.2 (10.1–15.0)	9.3 (6.4–13.0)	0.002
Acquired pneumonia	12 (10.1)	7 (7.3)	5 (21.7)	0.040
Acquired UTI	2 (1.7)	2 (2.1)	0 (0.0)	0.485
SOFA at admission	12.0 (11.0–15.0)	12.0 (11.0–15.0)	11.5 (10.5–15.0)	0.654
MV duration (day)	7.0 (3.0–9.0)	7.0 (2.0–9.0)	7.0 (5.0–13.0)	0.050

Data are presented as median with interquartile ranges or frequency with percentages. Abbreviations: ARDS = acute respiratory distress syndrome; BMI = body mass index; CA = cardiac arrest; MI = myocardial infarction; ACS = acute coronary syndrome; PCI = percutaneous coronary intervention; COPD = chronic obstructive pulmonary disorder; CKD = chronic kidney disease; LC = liver cirrhosis; CPR = cardiopulmonary resuscitation; NMB = neuromuscular blocking; ICU = intensive care unit; BT = body temperature; TTM = target temperature management; ROSC = return of spontaneous circulation; UTI = urinary tract infection; SOFA = sequential organ failure assessment; MV = mechanical ventilator.

**Table 2 jcm-09-00527-t002:** Comparisons of ABGA and mechanical ventilator parameters on admission in the ARDS patients.

Parameters	Total (n = 119)	Poor Outcome (n = 96)	Good Outcome (n = 23)	*p*-Value
pH	7.0 (6.9–7.2)	7.0 (6.8–7.2)	7.2 (7.0–7.3)	0.001
FiO_2_ (%)	100.0 (80.0–100.0)	100.0 (90.0–100.0)	100.0 (60.0–100.0)	0.028
PaO_2_ (mmHg)	86.0 (68.4–115.8)	86.0 (70.4–115.8)	77.1 (62.5–137.4)	0.287
PaCO_2_ (mmHg)	54.0 (40.0–73.0)	57.0 (43.5–76.0)	46.0 (32.6–60.5)	0.005
PEEP (cm H_2_O)	5.0 (4.0–8.0)	5.0 (4.0–8.0)	5.0 (4.0–7.0)	0.638
Inspiratory pressure (cm H_2_O)	15.0 (12.0–18.0)	16.0 (14.0–18.0)	12.0 (12.0–13.5)	<0.001
P_plat_ (cm H_2_O)	21.0 (17.3–25.8)	21.0 (18.0–22.0)	17.0 (16.0–20.0)	<0.001
Respiratory rate	20.0 (18.0–22.0)	20.0 (18.0–22.0)	18.0 (15.0–20.0)	0.013
Tidal volume (mL)	427.0 (367.0–497.0)	420.0 (360.0–489.5)	458.0 (395.5–545.0)	0.053
Compliance (ml/cm H_2_O)	29.7 (21.8–36.9)	27.5 (21.6–34.0)	38.6 (33.1–46.2)	<0.001
Minute ventilation (L/min)	8.2 (7.6–10.4)	8.8 (7.7–10.3)	8.8 (6.5–10.7)	0.609

Data are presented as median with interquartile ranges. Abbreviations: ABGA = arterial blood gas analysis; ARDS = acute respiratory distress syndrome; PEEP = positive end expiratory pressure; PIP = peak inspiratory pressure; P_plat_ = plateau pressure.

**Table 3 jcm-09-00527-t003:** Univariate and multivariate time-varying cox regression of lung compliance associated with favorable neurologic outcome.

Variables	Univariate			Multivariate		
	HR	95% CI	*p*-Value	Adjusted HR	95% CI	*p*-Value
Shockable	1.47	1.22–1.77	<0.001	1.28	1.01–1.63	0.039
Cardiac cause	2.10	0.92–4.82	0.080	1.99	0.55–7.23	0.293
Carbon dioxide	0.97	0.94–0.99	0.017	0.99	0.96–1.02	0.643
CPR duration	0.96	0.93–1.00	0.060	0.94	0.90–1.00	0.038
Compliance						
Maximum	1.03	1.01–1.05	<0.001	1.05	1.02–1.09	<0.001
Minimum	1.04	1.02–1.06	<0.001	1.08	1.03–1.13	<0.001
Median	1.04	1.02–1.06	<0.001	1.06	1.02–1.09	<0.001

Abbreviations: HR = hazard ratio; CI = confidence interval; CPR = cardiac pulmonary resuscitation.

**Table 4 jcm-09-00527-t004:** Performance of compliance to predict good outcomes.

	Sensitivity	Specificity	PLR	NLR	PPV	NPV
D1 max >32.5	69.5	72.0	2.48	0.42	90.4	38.3
D1 min >28.6	64.9	66.7	1.95	0.53	88.4	32.7
D1 median >31.4	59.3	66.7	1.78	0.61	90.0	24.5

Abbreviations: D = day; PLR = positive likelihood ratio; NLR = negative likelihood ratio; PPV = positive predictive value; NPV = negative predictive value.

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
