# Peer review of "Impact of Lung Compliance on Neurological Outcome in Patients with Acute Respiratory Distress Syndrome Following Out-of-Hospital Cardiac Arrest"

_jcm, 2020, doi:10.3390/jcm9020527_

Round 1

Reviewer 1 Report

Regarding the paper “Impact Of Lung Compliance On Neurological Outcome In Patients With Acute Respiratory Distress Syndrome Following Out-Of-Hospital Cardiac Arrest” submitted to JCM. I have the following comments:

The authors present their paper as a diagnostic paper were the prognostic ability of lung compliance appear to be a main purpose. So do the authors really think that this is relevant and a possible method? My impression is more that this paper may explore acute lung injury as one component of post cardiac arrest disease. If lung compliance is associated with higher mortality, what is the mechanism for this? The authors state that mechanical ventilation is a understudies feature of PCA care. This may the case but there are a few studies, one RCT looking at targeting different oxygen and carbon dioxide targets which in itself will results in very different TV, PEEP, FiO2 etc. (1). Why are the authors only looking at the subset of patients with ARDS? In addition what is ARDS in this context? I suspect that it is in most cases aspiration in combination with hypoxia. The authors should provide more data on based on what criteria patients were defined to have ARDS. A flow chart with detailed data on the excluded patients is needed. The authors discuss oxygen and hyperoxia as mechanism behind acute lung injury (beginning of the discussion). On what is this statement based? We have in two studies explored oxygen use during mechanical ventilation and its association with inflammation and neurological injury in TBI and cardiac arrest. Thus far the evidence is quite limited that oxygen is a main driver main the development of hypoxic brain injury and inflammation in these patients (2,3). Compliance is a component of TV, PEEP and peak pressure. Which one of these is important? Which one of these can be modified? It would be useful to look closer at whether it is peak pressure, PEEP or TV which has the stronger association with outcome. This could be done my comparing the AUC of these. Did patients receive a paralysis? Major point and it will influence compliance to a great degree. Why is the mean, max and median compliance so important? It is likely that if compliance is measured when the patient is “waking up” and/or shivering compliance will change and a measurement during this time is not likely to be relevant. How about oxygen and carbon dioxide overall? Was carbon dioxide included in the multivariate model? Relevant as this will be a result on MV setting and has been shown to be associated with outcome in several studies (4-5).

Jakkula P, Pettila V, Skrifvars MB, Hästbacka J, Loisa P, Tiainen M, Wilkman E, Toppila J, Koskue T, Bendel S, Birkelund T, Laru-Sompa R, Valkonen M, Reinikainen M for the COMACARE study group. Targeting low-normal or high-normal mean arterial pressure after cardiac arrest and resuscitation: a randomised pilot trial. Intensive Care Medicine 2018;44:2091-2101. Nelskyla A,Nurmi J, Jousi M, Schramko A, Mervaala E, Ristagno G, Skrifvars MB. The effect of 50% compared to 100% inspired oxygen fraction on brain oxygenation and post cardiac arrest mitochondrial function in experimental cardiac arrest. Resuscitation. 2017 Jul;116:1-7. Lang M, Skrifvars MB, Siironen J, Tanskanen P, Alapeijari M, Koivisto T, Bendel S. The effect of moderate hyperoxemia on neurological injury, inflammation and oxidative stress – A randomised controlled pilot trial (BRAINOXY NCT01201291). Acta Anaesthesiol Scand. 2018 Jul;62(6):801-810. Vaahersalo J, Bendel S, Reinikainen M, Kurola J, Tiainen M, Raj R, Pettila V, Varpula T, Skrifvars MB, FINNRESUSCI Investigators. Arterial blood gas tensions after resuscitation from out-of-hospital cardiac arrest: associations with long-term neurological outcome. Critical Care Medicine 2014 Jun;42(6):1463-70. Eastwood GM, Schneider AG, Suzuki S, Peck L, Young H, Tanaka A, Mårtensson J, Warrillow S, McGuinness S, Parke R, Gilder E, Mccarthy L, Galt P, Taori G, Eliott S, Lamac T, Bailey M, Harley N, Barge D, Hodgson CL, Morganti-Kossmann MC, Pébay A, Conquest A, Archer JS, Bernard S, Stub D, Hart GK, Bellomo R. Targeted therapeutic mild hypercapnia after cardiac arrest: A phase II multi-centre randomised controlled trial (the CCC trial). Resuscitation. 2016 Jul;104:83-90.

Author Response

Reviewer 1

Comments and Suggestions for Authors

Regarding the paper “Impact Of Lung Compliance On Neurological Outcome In Patients With Acute Respiratory Distress Syndrome Following Out-Of-Hospital Cardiac Arrest” submitted to JCM. I have the following comments:

The authors present their paper as a diagnostic paper were the prognostic ability of lung compliance appear to be a main purpose. So do the authors really think that this is relevant and a possible method? My impression is more that this paper may explore acute lung injury as one component of post cardiac arrest disease.

Response>

Thank you for your comment. A multimodal diagnostic approach for the prediction of outcome in patients with cardiac arrest is currently used to minimize prognostic uncertainty. However, one of the most pressing issues for relatives and healthcare workers is to rapidly obtain reliable information regarding the probability of achieving good neurological outcomes. Although numerous studies have focused on discovering factors for poor neurologic outcome, it is essential to develop strategies for predicting good neurologic outcomes among OHCA survivors in order to appropriately tailor medical therapies for each patient.

We have an experience that ARDS/acute lung injury is commonly observed in PCAS patients, but in some patients, clinical features are different from those of conventional ARDS. They recovered from the ARDS and discharge with good neurologic outcome. It appears that in ARDS patients from cardiac arrest were not homogenous population and needed risk stratification. Therefore, we tried to evaluated the which ventilator parameters is associated with neurological outcome and then we found lung compliance. However, we agreed with reviewer’s concern about lung compliance which may have affected by many situations. We just reported here the possibility of lung compliance as an early prognostic tool in cardiac arrest survivors. Further studies will be needed.

If lung compliance is associated with higher mortality, what is the mechanism for this?

Response>

Whether ventilator parameters including lung compliance are causally related to outcome or whether they are simple markers of disease severity is unclear. In our ICU, all PCAS patients received lung protective ventilatory strategy. In this setting, lung compliance may be more informative than other ventilatory parameters such as PEEP, TV, and peak airway pressure. Poor outcome group had lower lung compliance in our study and it could be explained by multifactorial reasons. First of all, lung compliance could reflect the severity of the ARDS. Numerous reports announced that patients with severe ARDS itself had higher mortality in critically ill patients not only in patients with PCAS. Moreover, severe ARDS aggravated the unbalance levels of oxygen and carbon dioxide which could impair recovering neurocognitive function. Physicians tend to stop treatment PCAS patients, and this decision is unrelated with medical conditions. Lastly, patients with lower lung compliance could be less tolerable from ventilator induced lung injury like barotrauma. We discussed these in the discussion section of the manuscript.

“We identified the correlations between lung compliances and neurologic outcomes. This could be explained by multifactorial reasons. First of all, lung compliance could reflect the severity of the ARDS. Numerous reports announced that patients with severe ARDS itself had higher mortality in critically ill patients not only in patients with PCAS [12]. Moreover, severe ARDS aggravated the unbalance levels of oxygen and carbon dioxide which could impair recovering neurocognitive function.” (line 254-259)

“Lastly, patients with lower lung compliance could be less tolerable from ventilator induced lung injury like barotrauma [26].” (line 266-267)

The authors state that mechanical ventilation is a understudies feature of PCA care. This may the case but there are a few studies, one RCT looking at targeting different oxygen and carbon dioxide targets which in itself will results in very different TV, PEEP, FiO2 etc. (1). Why are the authors only looking at the subset of patients with ARDS?

Response>

Postcardiac arrest syndrome can affect multi-organ injury and pulmonary involvement manifests as ARDS. We think that preventing further ventilatory-induced lung injury and managing adequate level of oxygen and carbon dioxide seemed to be more important in PCAS patients with ARDS than without ARDS. We already found that the occurrence of ARDS in PCAS was associated with poor outcome (not published) and try to find markers for stratifying risk.

In addition what is ARDS in this context? I suspect that it is in most cases aspiration in combination with hypoxia. The authors should provide more data on based on what criteria patients were defined to have ARDS.

Response>

The mechanisms leading to acute lung injury/ARDS in the postcardiac arrest syndrome are neuro-humoral, immune-mediated and iatrogenic etiologies [1]. Among iatrogenic factor, chest compression induced lung contusion is not preventable but oxygen toxicity and ventilator-induced lung injury should be prevented during admission. Development of ARDS was identified by the result of arterial blood gases and chest images according to the Berlin definition.

Moreover, for diagnosing more definitely, we additionally found detail traditional risk factors of ARDS in our study population and provided detail lists in Table 1.

Traditional ARDS risk factors

Shock: 69 (58.0%)

Aspiration: 62 (52.1%)

Pancreatitis: 3 (2.5%)

Pneumonia: 44 (37.0%)

Pulmonary contusion: 85 (71.4%)

Sepsis, no pneumonia: 19 (16.0%)

“In brief, we found those who had classic risk factors of the development of ARDS by reviewing medical charts, had two consecutive arterial blood gas analysis (ABGA) with partial pressure of arterial oxygen (PaO2)/fraction of inspired oxygen (FiO2) (P/F) ratio < 300 mm Hg in a mechanically ventilated patient, had bilateral radiographic opacities on chest radiography or on computed tomography that could not be explained by effusions, pulmonary collapse, or nodules, and had respiratory failure that could not be explained by cariogenic pulmonary edema. All chest images were reviewed by radiologists.” (line 74-77)

“Based on the results of the chest images, laboratory data, and needs of vasopressors, the traditional risk factors of developing ARDS, such as shock, aspiration, pancreatitis, pneumonia, pulmonary contusion, and sepsis were recorded by physicians on duty.” (line 119-122)

A flow chart with detailed data on the excluded patients is needed.

Response>

We provided a flow chart with detail number of patients in each step.

“During the study period, a total of 295 registry patients survived CA, and 246 were analyzed after excluding 49 who received ECMO and had cardiogenic pulmonary edema (Fig. 1).” (line 144-145)

The authors discuss oxygen and hyperoxia as mechanism behind acute lung injury (beginning of the discussion). On what is this statement based? We have in two studies explored oxygen use during mechanical ventilation and its association with inflammation and neurological injury in TBI and cardiac arrest. Thus far the evidence is quite limited that oxygen is a main driver main the development of hypoxic brain injury and inflammation in these patients (2,3).

Response>

Several animal studies indicate that hyperoxaemia early after return of spontaneous circulation causes oxidative stress and harms post-ischaemicneurons [2-4]. Landmark study that included more than 6000 patients supports the animal data and shows post-resuscitation hyperoxaemia in the first 24 h is associated with worse outcome, compared with both normoxaemia and hypoxaemia [5]. A further analysis by the same group showed that the association between hyperoxia and outcome was dose-dependent and that there was not a single threshold for harm [6]. An observational study that included only those patients treated with mild induced hypothermia also showed an association between hyperoxia and poor outcome [7]. So current CPR guidelines for PCAS patients suggest titrate the FiO2 to maintain the SaO2 in the range of 94–98% to avoid hypoxaemia [8]. We added these evidences and references in discussion section.

“Moreover, severe ARDS aggravated the unbalance levels of oxygen and carbon dioxide which could impair recovering neurocognitive function. Several animal studies indicate that hyperoxaemia early after return of spontaneous circulation causes oxidative stress and harms post-ischaemic neurons [25-27]. Landmark study that included more than 6000 patients supports the animal data and shows post-resuscitation hyperoxaemia in the first 24 h is associated with worse outcome, compared with both normoxaemia and hypoxaemia [28]. A further analysis by the same group showed that the association between hyperoxia and outcome was dose-dependent and that there was not a single threshold for harm [30]. An observational study that included only those patients treated with mild induced hypothermia also showed an association between hyperoxia and poor outcome [31].” (line 259-267)

Compliance is a component of TV, PEEP and peak pressure. Which one of these is important? Which one of these can be modified? It would be useful to look closer at whether it is peak pressure, PEEP or TV which has the stronger association with outcome. This could be done my comparing the AUC of these.

Response>

Thank you for your suggestion. In Figure 3, we calculated ROC curves of compliances with TV, PEEP, and PIP and compared AUC. All of our patients were set with pressure-control mode, and both PEEP and inspiratory pressure were modified by physicians. Meanwhile, TV and compliance were monitored variables. Comparing AUC of these parameters was presented in Figure 3. All day-1 maximum, minimum, and median compliances showed higher AUC than that of TV.

Figure 3. ROC comparisons of maximum, minimum, and median day-1 compliance with PEEP, PPlat, and VT. A. Comparison of day-1 max, medium, and minimum compliance. B. Comparison of day-1 maximum PEEP, VT, Pplat, compliance. C. Comparison of day-1 minimum PEEP, Pplat, VT, and compliance. D. Comparison of day-1 median PEEP, Pplat, VT, and compliance. Abbreviations: PEEP = positive end expiratory pressure; Pplat = plateau pressure; VT = tidal volume.

Did patients receive a paralysis? Major point and it will influence compliance to a great degree.

Response>

We totally agreed with your opinion that neuromuscular agent might be influence the lung compliance. In our ICU, we used neuromuscular agent between return of spontaneous circulation and get a target temperature (usually 33’C) to reduce shivering, and keep synchronizing respiration of patients with ventilator. And then the use of neuromuscular agent depends on the treating physician. They usually use neuromuscular agent for severe ARDS patients.

We corrected additional variables for those who got continuous neuromuscular blocking agents, and there was no difference between poor and good outcome groups (43.8% vs 52.2%, p = 0.467) (Table1).

“Neuromuscular blocking agents were used from return of spontaneous circulation and achieved target temperature to reduce shivering and keep synchronizing respiration of patients with ventilator.” (line 95-97)

Why is the mean, max and median compliance so important? It is likely that if compliance is measured when the patient is “waking up” and/or shivering compliance will change and a measurement during this time is not likely to be relevant.

Response>

Early predicting good neurologic outcomes among OHCA survivors is important to appropriately tailor medical therapies for each patient. We agreed that first 24hr is not be relevant, but we were to evaluate compliance as a early prognostic tool.  Lung compliances will change during first 24hr. Each of max, minimum, and median compliance could reflect the best, worst, mean condition of lung elasticity.

“We supposed that lung compliances were changing dynamically during intensive care. Each of max, minimum, and median compliance could reflect the best, worst, mean condition of lung elasticity. We tried to evaluate which conditions were associated with clinical outcomes.” (line 269-272)

How about oxygen and carbon dioxide overall?

Response>

We added overall oxygen and carbon dioxide levels during a observation period in Table S1 (Additional file).

Characteristics

Total

(n = 119)

Poor outcome

(n = 96)

Good outcome

(n = 23)

p-value

PaO2 (mmHg)

  After ROSC

86.0 (68.4 – 115.8)

86.0 (70.4 – 115.8)

77.1 (62.5 – 137.4)

0.287

D1 maximum

113.0 (88.0 – 143.0)

109.0 (94.0 – 131.9)

139.5 (107.1 – 156.8)

0.374

D1 minimum

69.0 (57.5 – 83.9)

64.0 (58.0 – 77.0)

73.5 (63.2 – 82.5)

0.230

D1 median

88.3 (72.0 – 110.5)

83.3 (69.0 – 95.0)

102.3 (94.9 – 114.2)

0.448

D2 maximum

102.9 (86.0 – 133.3)

106.0 (90.8 – 143.0)

109.7 (90.8 – 124.4)

0.418

D2 minimum

69.0 (60.0 – 78.8)

72.5 (64.0 – 79.6)

80.8 (64.8 – 93.4)

0.097

D2 median

81.0 (71.4 – 91.3)

80.0 (72.0 – 88.0)

90.8 (82.1 – 101.5)

0.216

D3 maximum

97.1 (84.3 – 108.0)

101.0 (81.0 – 110.0)

97.1 (83.0 – 114.0)

0.807

D3 minimum

72.0 (60.0 – 78.0)

70.0 (58.0 – 79.7)

76.1 (61.8 – 81.5)

0.120

D3 median

82.8 (69.0 – 90.8)

79.4 (65.0 – 91.2)

85.9 (71.8 – 94.6)

0.085

D4

87.6 (74.0 – 92.3)

87.0 (74.0 – 95.0)

95.5 (87.4 – 102.2)

0.417

D5

90.4 (70.0 – 101.0)

88.5 (70.0 – 101.0)

95.5 (88.9 – 102.6)

0.263

  D6

81.0 (70.8 – 96.0)

81.0 (69.0 – 95.8)

83.9 (75.0 – 137.4)

0.660

PaCO2 (mmHg)

  After ROSC

54.0 (40.0 – 73.0)

57.0 (43.5 – 76.0)

46.0 (32.6 – 60.5)

0.005

D1 maximum

40.5 (34.4 – 49.1)

43.4 (39.0 – 52.0)

38.3 (32.6 – 44.3)

0.027

D1 minimum

44.0 (37.0 – 52.0)

44.1 (39.0 – 53.3)

46.5 (41.9 – 55.1)

0.950

D1 median

40.9 (35.7 – 47.4)

42.0 (38.0 – 46.0)

43.7 (33.8 – 47.6)

0.362

D2 maximum

37.8 (33.0 – 42.0)

41.0 (33.0 – 44.8)

40.5 (34.8 – 44.6)

0.085

D2 minimum

41.0 (35.5 – 47.5)

38.2 (35.7 – 46.0)

41.0 (35.7 – 46.9)

0.216

D2 median

40.8 (36.0 – 46.8)

43.0 (36.0 – 47.0)

50.0 (42.4 – 53.4)

0.304

D3 maximum

39.0 (35.0 – 42.9)

38.0 (35.0 – 41.0)

39.5 (33.6 – 43.7)

0.510

D3 minimum

41.5 (38.4 – 46.0)

43.0 (39.0 – 46.0)

41.8 (38.9 – 47.7)

0.932

D3 median

41.7 (38.0 – 44.5)

41.8 (38.0 – 45.2)

41.4 (41.2 – 44.5)

0.137

D4

40.6 (36.8 – 45.9)

41.0 (36.9 – 48.0)

43.3 (37.4 – 47.7)

0.925

D5

40.0 (35.1 – 45.9)

40.0 (32.8 – 49.0)

37.0 (34.5 – 43.4)

0.336

D6

39.1 (33.4 – 44.0)

40.0 (34.0 – 48.0)

33.6 (31.5 – 43.1)

0.298

Data are presented as median with interquartile ranges.

Abbreviations: ARDS = acute respiratory distress syndrome; ROSC = return of spontaneous circulation; D = day.

Was carbon dioxide included in the multivariate model? Relevant as this will be a result on MV setting and has been shown to be associated with outcome in several studies (4-5).

Response>

Thank you for your suggestion. Even though carbon dioxide significantly different between groups, adjusted model for multivariate model showed that the level of carbon dioxide was not an independent factor for predicting outcomes (adjusted HR 0.99, 95% CI [0.96 – 1.02], p = 0.643). We added this results in table 3.

Table 3. Univariate and multivariate time-varying cox regression of lung compliance associated with favorable neurologic outcome

Variables

Univariate

Multivariate

HR

95% CI

p-value

Adjusted HR

95% CI

p-value

Shockable

1.47

1.22 – 1.77

< 0.001

1.28

1.01 – 1.63

0.039

Cardiac cause

2.10

0.92 – 4.82

0.080

1.99

0.55 – 7.23

0.293

Carbon dioxide

0.97

0.94 – 0.99

0.017

0.99

0.96 – 1.02

0.643

CPR duration

0.96

0.93 – 1.00

0.060

0.94

0.90 – 1.00

0.038

Compliance

Maximum

1.03

1.01 – 1.05

< 0.001

1.05

1.02 – 1.09

< 0.001

Minimum

1.04

1.02 – 1.06

< 0.001

1.08

1.03 – 1.13

< 0.001

Median

1.04

1.02 – 1.06

< 0.001

1.06

1.02 – 1.09

< 0.001

Abbreviations: HR = hazard ratio; CI = confidence interval; CPR = cardiac pulmonary resuscitation.

Reference

(The post-cardiac arrest syndrome: A case for lung–brain coupling and opportunities for neuroprotection, Journal of Cerebral Blood Flow & Metabolism 39(6):0271678X1983555· March 2019) Sutherasan, Y.; Penuelas, O.; Muriel, A.; Vargas, M.; Frutos-Vivar, F.; Brunetti, I.; Raymondos, K.; D’Antini, D.; Nielsen, N.; Ferguson, N.D.; Bottiger, B.W.; Thiller, A.W.; Davies, A.R.; Hurtado, J.; Rios, F.; Apeztegula, C.; Violi, D.A.; Caker, N.; Gonzalez, M.; Du, B.; Kuiper, M.A.; Soares, M.A.; Koh, Y.; Moreno, R.P.; Amin, P.; Tomicic, V.; Soto, L.; Bulow, H.H.; Anzueto, A.; Esteban, A.; Pelosi, P. for VENTILA GROUP. Management and outcome of mechanically ventilated patients after cardiac arrest. Crit Care 2015, 19, 215-226. Pilcher, J.; Weatherall, M.; Shircliffe, P.; Bellomo, R.; Young, P.; Beasley, R. The effect of hyperoxia following cardiac arrest – a systematic review and meta-analysis of animal trials. Resuscitation 2012, 83, 417–422. Richards, E.M.; Fiskum, G.; Rosenthal, R.E.; Hopkins, I.; McKenna, M.C. Hyperoxic reperfusion after global ischemia decreases hippocampal energy metabolism. Stroke 2007, 38, 1578–1584. Kilgannon, J.H.; Jones, A.E.; Shapiro, N.I.; Angelos, M.G.; Milcarek, B.; Hunter, K.; Parrillo, J.E.; Trzeciak, S. for Emergency Medicine Shock Research Network (EMShockNet) Investigators. Association between arterial hyperoxia following resuscitation from cardiac arrest and in-hospital mortality. JAMA 2010, 303, 2165–2171. Kilgannon, J.H.; Jones, A.E.; Parrillo, J.E.; Dellinger, R.P.; Milcarek, B.; Hunter, K.; Shapiro, N.I.; Trzeciak, S. for Emergency Medicine Shock Research Network (EMShockNet) Investigators. Relationship between supranormal oxygen tension and outcome after resuscitation from cardiac arrest. Circulation 2011, 123, 2717–2722. Janz, D.R.; Hollenbeck, R.D.; Pollock, J.S.; McPherson, J.A.; Rice, T.W. Hyperoxia is associated with increased mortality in patients treated with mild therapeutic hypothermia after sudden cardiac arrest. Crit Care Med 2012, 40, 3135–3139. Nolan, J.P.; Soar, J.; Cariou, A.; Croberg, T.; Moulaert, V.R.; Deakin, C.D.; Bottiger, B.W.; Friberg, H.; Sunda, K.; Sandroni, C. European Resuscitation Council and European Society of Intensive Care Medicine Guidelines for Post-resuscitation Care 2015Section 5 of the European, Resuscitation Council Guidelines for Resuscitation 2015. Resuscitation 2015, 202–222.

Reviewer 2 Report

Introduction:

line 42-43: needs citations

Material & Methods:

section 2.2:

add average time of CA to resuscitation (ALS)

add standard protocol of ALS (vasopressors, amiodarone)  in pre-clinic setting, add sedation protocols used on ICU

add average levels of temperature before TTM

line 97: define why you have chosen such a short observation period of 7 days. Discuss potential bias due to excluding potential confounders due to longer observation period

line 98: You should exclude patients with incomplete data.

Results:

line 125: define ARDS severity in the M&M section prior

Table 1: add BMI and COPD, history of smoking 

add a table with possible confounders (related due to treatment on ICU) such as pneumonia, urinary tract infection, sedation medication etc.

Why did you not not consider TTM and pre TTM temperature levels as potential source of influence on neurological outcome?

Table 2: add lactate levels after ROSC

Discussion:

line 234-238:

Here you mention major issues concerning the reliability of you results. You should include these potential confounders in your study protocol and calculations

Conclusion:

Line 240-241: This statement might be right according to your very specific setting but might be not applicable in patients with longer observation period and regarding circumstances like pneumonia, infection, COPD, time from CA to resuscitation 

Author Response

Reviewer 2

Comments and Suggestions for Authors

Introduction:

line 42-43: needs citations

Response>

We added citations.

“The authors postulated that ARDS accelerates ischemic brain injury caused by aggravating the mismatch of supply and demand of oxygen (O2) and carbon dioxide (CO2) [5].” (line 43-45)

Material & Methods:

section 2.2:

add average time of CA to resuscitation (ALS)

Response>

We added average time of CA to resuscitation (ALS) in table 1. There were no differences between groups (5.0 min vs 4.0 min, p = 0.708) (Table 1). We described in material and method section.

“In addition, we gathered other CA data according to the Utstein Style recommendations, including location, witnessed arrest, bystander CPR, time gap between perception of cardiac arrest and advanced cardiac life support, duration of CPR, initial documented rhythm, presumed arrest cause, and initial electrocardiography rhythm.”  (line 116-119)

add standard protocol of ALS (vasopressors, amiodarone)  in pre-clinic setting

Response>

Emergency medical services in South Korea did not use vasopressors or amiodarone before ED arrival (Scoop and run strategy) except some special cases. We reviewed and found only two cases of poor outcome group in the study population that use epinephrine before ED arrival. Therefore, it would be okay to omit data of pre-clinic medications in the analysis.

add sedation protocols used on ICU

Response>

Patients were deeply sedated (defined by a score on the Richmond Agitation-Sedation Scale of -2-  to -4) for preventing shivering or avoiding awareness during TTM. After finishing rewarming phase, light sedation (defined by a score on the Richmond Agitation-Sedation Scale of 0 or -1) was warranted. We added these contents to 2.2. Routine post-cardiac arrest care protocol section. In addition, we provided data about the various type of sedative agents for patients in Table 1.

“Patients were deeply sedated (defined by a score on the Richmond Agitation-Sedation Scale of -2-  to -4) for preventing shivering or avoiding awareness during TTM. After finishing rewarming phase, light sedation (defined by a score on the Richmond Agitation-Sedation Scale of 0 or -1) was warranted.” (line 97-100)

Table 1. Baseline characteristics of the ARDS patients

Characteristics

Total

(n = 119)

Poor outcome

(n = 96)

Good outcome

(n = 23)

p-value

Sedative and NMB during TTM

Midazolam

26 (21.8)

22 (22.9)

4 (17.4)

0.565

Ketamine

2 (1.7)

2 (2.1)

0 (0.0)

0.352

Morphine

38 (31.9)

31 (32.3)

7 (30.4)

0.863

Propofol

80 (67.2)

59 (61.5)

21 (91.3)

0.003

Fentanyl

58 (48.7)

45 (46.9)

13 (56.5)

0.405

Abbreviations: ARDS = acute respiratory distress syndrome; NMB = neuromuscular blocking; TTM = therapeutic target temperature management

add average levels of temperature before TTM

Response>

The data of the average levels of temperature before TTM were inserted in table 1 and mentioned in 2. Material and Method sections.

“Temperature was recorded before starting TTM, and monitored using an esophageal probe and continuous intravenous propofol and opioids (morphine or remifentanil) were administered for sedation and analgesia during TTM.” (line 93-95)

line 97: define why you have chosen such a short observation period of 7 days. Discuss potential bias due to excluding potential confounders due to longer observation period

Response>

Neurologic prognostication was routinely performed after 72 hours from ROSC based on current CPR guideline. Depends on the neurocognitive status, physicians tend to consider whether keep conservative medical treatment or not. Moreover, measuring compliance is impossible when patients extubated or death (only 30 patients with mechanical ventilation at 28days in our population). These are the reason why we chosen a short observation period of 7 days. We agreed with reviewer’s concern and we discussed this as limitations on the manuscript.

“Fifth, because of relatively small sample size, we could collect only 7-days follow-up data. Measuring compliance is impossible when patients extubated or death. When we follow-up our study population, ARDS patients were 119 on day-1, but remained only 30 patients with mechanical ventilation. Future study with bigger sample size may reveal long-term clinical impacts of ARDS on patients with PCAS.” (line 297-301)

line 98: You should exclude patients with incomplete data.

Response>

There were 9 patients those who performed only 2 ABGAs on hospital day-1 because they admitted ED just before the end of the day. We calculated compliances and divided into max and minimum without median. Because there was relatively small sample size, we included 9 patients with filling mean values for missing median values. However, we totally agreed with your concern that this method could influence the result. Therefore, we additionally performed sensitivity analysis with abstracting these 9 patients, and confirmed that the trends of results were not changed.

“Sixth, 9 patients performed only 2 ABGAs on hospital day-1 because they admitted ED just before the end of the day. We calculated compliances and divided into max and minimum without median. Because there was relatively small sample size, we included 9 patients with filling mean values for missing median values. In order to make sure that these missing variables would not influence the results, we additionally performed sensitivity analysis with abstracting these 9 patients, and confirmed that the trends of results were not changed.” (line 301-307)

Results:

line 125: define ARDS severity in the M&M section prior

Response>

We provided the detail definition of the severity of ARDS in the 2. Materials and Methods section.

“The severity of the ARDS was determined based on degree of hypoxemia: mild (200 mmHg < PF ratio ≤ 300 mmHg), moderate (100 mmHg < PF ratio ≤ 200 mmHg), and severe (PF ratio < 100 mmHg).” (line 81-82)

Table 1: add BMI and COPD, history of smoking 

Response>

We added BMI, COPD, and history of smoking in table 1, and described in the manuscript. All of three variables were similar between two groups of patients.

“Body mass index and the proportion of smoker were not different between two groups.” (line 150-151)

Table 1. Baseline characteristics of the ARDS patients

Characteristics

Total

(n = 119)

Poor outcome

(n = 96)

Good outcome

(n = 23)

p-value

BMI (kg/m2)

23.7 (21.1 – 26.1)

22.9 (20.8 – 26.0)

23.3 (18.9 – 26.2)

0.719

Smoker

44 (37.0)

33 (34.4)

11 (47.8)

0.235

COPD

12 (10.1%)

12 (12.5)

0 (0.0)

0.075

Abbreviations: ARDS = acute respiratory distress syndrome; BMI = body mass index; COPD = chronic obstructive pulmonary disorder.

add a table with possible confounders (related due to treatment on ICU) such as pneumonia, urinary tract infection, sedation medication etc.

Response>

According to your suggestion, we added possible confounders related to ICU care in the table 1, including hospital-acquired pneumonia, urinary tract infection, and sedative agents. Except Propofol and acquired pneumonia, there were no differences of ICU care factors between two groups. Propofol was used more often in the patients with good neurologic outcome group than that of poor outcome. We mentioned these in the manuscript.

“Besides of propofol, all of sedative and neuromuscular blocking agents were used similarly between both groups. Initial lactate levels after ROSC were significantly higher in patients with poor neurologic outcome (13.2 mmol/L vs 9.3 mmol/L, p = 0.002).” (line 159-161)

Why did you not consider TTM and pre TTM temperature levels as potential source of influence on neurological outcome?

Response>

We agreed your concern that pre TTM or TTM temperature levels might influence the recovery of neurological outcome. We collected pre TTM temperature and inserted data in table 1. In case of TTM temperature, our study facility have preferred to set the TTM temperature as 33 ℃, however, some cases were set as 36 if there were severe coagulopathy, non-sustained shock, or extreme bradycardia. In this study, total inclusion patients with ARDS were set with 33 ℃. Therefore, we added and corrected sentences in the “2.2. Routine post-cardiac arrest care protocol” to reduce confusion.

“The target temperature of 33℃ was maintained for 24 hours (maintenance phase), and patients were then rewarmed to 36.5℃ at a rate of 0.25℃/hr.” (line 92-93)

“Both groups showed similar body temperature before starting TTM, and all patients had TTM with 33℃ during maintenance.” (line 157-158)

Table 2: add lactate levels after ROSC

Response>

Thank you for your suggestion, and the initial lactate levels after ROSC were inserted in table 1. Lactate level was significantly higher in the poor outcome group. This result was correlated with more frequently developed shock in poor outcome group. We stated these results in the result and method section.

“Initial lactate level just after ROSC was performed in every survivor.” (line 101-102)

“Initial lactate levels after ROSC were significantly higher in patients with poor neurologic outcome (13.2 mmol/L vs 9.3 mmol/L, p = 0.002).” (line 160-161)

Discussion:

line 234-238:

Here you mention major issues concerning the reliability of you results. You should include these potential confounders in your study protocol and calculations

Response>

We totally agree with your opinion that it is hard to exclude all cardiogenic pulmonary edema which decrease lung compliance without true ARDS. Previous reports announced that postcardiac arrest myocardial dysfunction was common and the most common type was LV systolic dysfunction. Therefore, we used our echocardiographic data and exclude 9 patients those who had severe LV dysfunction (LV ejection fraction < 30%). But, the true incidence of cardiogenic respiratory failure remains unclear. To solve this problem, we performed a sensitivity analysis without excluding cardiac failure and get similar trend of the study result that the compliance is an independent risk factors for predicting neurologic outcomes of survivors from cardiac arrest. We added these contents in the manuscript.

“To strengthen our result, we additionally performed a sensitivity analysis without excluding cardiac failure. Although specific values of HR changed, we found similar trend of the results that compliance was an independent risk factors for predicting neurologic outcomes [see Additional file.  Table S2].” (line 291-294)

Conclusion:

Line 240-241: This statement might be right according to your very specific setting but might be not applicable in patients with longer observation period and regarding circumstances like pneumonia, infection, COPD, time from CA to resuscitation 

Response>

We totally accept your concerns that longer observation period and other confounders make our conclusions different. However, in this study, I’d just like to find the ventilator parameters as early prognostic tool in PCAS patients and find a compliance first 24hr was associated with good neurological outcome.  To robust our results, we additionally collected potential confounding variables, including risk factors of ARDS, history of COPD, time gap between arrest and resuscitation (ACLS). We added these parameters in table 1, and found that COPD, hospital-acquired pneumonia, combined infections, and time gap did not differ between groups. Meanwhile, it is not certain that longer observation period might affect our results. We added our explanation to clarify our insisting in the manuscript.

“Fifth, because of relatively small sample size, we could collect only 7-days follow-up data. Measuring compliance is impossible when patients extubated or death. When we follow-up our study population, ARDS patients were 119 on day-1, but remained only 30 patients with mechanical ventilation on day-7. Future study with bigger sample size may reveal long-term clinical impacts of ARDS on patients with PCAS.” (line 297-298)

Reviewer 3 Report

Summary:

The authors describe a retrospective study evaluating the association between pulmonary compliance and neurologic outcomes in adults who have experienced a cardiac arrest. The authors find that poor pulmonary compliance on day 1 is associated with poor long-term neurologic outcome.  

Comments:

This is an interesting paper with important findings. However, the paper could be improved by reporting of additional data, improving the presentation of data, and augmenting the discussion.

Introduction

The second paragraph of the introduction states, “Therefore, discovering accurate prognostic markers of ARDS severity…” The authors later state that the goal in this study is to determine the association between lung compliance and neurologic outcome. These statements seem to conflict and leads to confusion as to the goal of the study. The authors describe that NMB agents improve outcomes in ARDS. This has recently been refuted in the ROSE study.

Methods:

Please provide a definition that describes CPC score of 1-2 and how it was assessed (chart review, direct assessment by trained providers, etc). The authors state that secondary outcomes include ARDS vs non-ARDS patient outcomes. This data is not presented.

Results:

Please add FiO2 values to Table 2. It would be helpful to have more complete ventilator data presented in the supplement, similar to the data presented in Table 2.  

Discussion:

Paragraph one of the discussion could be improved by adding a brief summary of the predictive modeling results. The majority of the discussion is related to incidence of ARDS after OHCA which was not the objective of the paper. Consider revising. The discussion could be improved by a brief discussion related to the low NPV of the test. The authors discuss change over 7 days (line 220) but do not show these results. The authors state “These results imply that the recovery of lung and brain injury is determined in the initial phase…” While the pulmonary compliance may improve over the initial few days, the authors do not present any evidence related to brain injury recovery. This statement should be modified. The limitations section of the paper should reiterate the retrospective design and inability to differentiate association from causation.

Figure 1-

The text reference to figure 1 states: “For all seven compliances, day 1 showed significant large differences between the good and poor outcome groups that decreased over 7 days.” The figure does not show change over 7 days. This statement is confusing to me. Please label axes and legend.

Minor comments:

Line 66: OHCA acronym incorrect. Line 98: Authors state “missing data…means of each variable.” Please clarify whether this is patient means or population means. Also, the number of missing data points should be reported in the results. Line 163: missing the word “most” Add units to compliance measurements consistently throughout the manuscript.

Author Response

Reviewer 3

Comments and Suggestions for Authors

Summary: The authors describe a retrospective study evaluating the association between pulmonary compliance and neurologic outcomes in adults who have experienced a cardiac arrest. The authors find that poor pulmonary compliance on day 1 is associated with poor long-term neurologic outcome.  

Comments: This is an interesting paper with important findings. However, the paper could be improved by reporting of additional data, improving the presentation of data, and augmenting the discussion.

Introduction

The second paragraph of the introduction states, “Therefore, discovering accurate prognostic markers of ARDS severity…” The authors later state that the goal in this study is to determine the association between lung compliance and neurologic outcome. These statements seem to conflict and leads to confusion as to the goal of the study.

Response>

Thank you for your suggestion. We accept your opinion that introduction section making confuse to readers about primary purpose of the study. We have an experience that ARDS is commonly observed in PCAS patients, but in some patients, clinical features are different from those of conventional ARDS. They recovered from the ARDS and discharge with good neurologic outcome. It appears that in ARDS patients from cardiac arrest were not homogenous population and needed risk stratification. Therefore, we tried to evaluated the which ventilator parameters is associated with neurological outcome and then we hypothesized lung compliance will have a prognostic value than TV or PEEP because these patients were treated lung-protective ventilatory strategy. We added some sentences to make clear the study purpose.

“Moreover, recent study introduced that lung could modulate the neurological injury, and recovering from ARDS could reduce brain damage [11]. Therefore, discovering accurate prognostic markers of ARDS severity is essential to facilitate risk stratification and apply novel therapeutic interventions to improve outcomes and predict good neurologic outcomes among CA survivors. Since lung compliance correlates with the aerated functional component of the pulmonary system [12], we hypothesized that lung compliance, an easily measured clinical variable, could be an important prognostic marker than VT or PEEP in patients with ARDS after CA.” (line 52-58)

The authors describe that NMB agents improve outcomes in ARDS. This has recently been refuted in the ROSE study.

Response>

The usefulness of early NMB infusion in moderate to severe ARDS patients have been controversial. The ROSE study enrolled patients of moderate-to-severe ARDS and found there were no different 90-day mortality. This result was different from prior ACURASYS study that concluded early NMB improved 90-day survival rate. Different results might be caused by various hidden confounding factors, such as PEEP strategy, sedation target, and prone position effect. Although applying NMB have been controversial, the guideline still suggested that applying NMB to severe ARDS (ie, PF ratio < 150). However, in order to reduce controversial sentence, we corrected sentence in 1. Introduction as follow.

“Even though the benefit of interventions has been controversial, guideline have suggested that fluid restriction, use of neuromuscular blocking agents, and prone ventilation could reduce mortality in severe ARDS patients [10].” (line 49-52)

Methods:

Please provide a definition that describes CPC score of 1-2 and how it was assessed (chart review, direct assessment by trained providers, etc).

Response>

Thank you for your kindness. We added sentences about the detail CPC score of 1-2 and the method how we collected data of CPC score in the “2.3. Definition of variables” section.

“The CPC score ranges from 1 to 5 with 1 presenting normal brain function and 5 presenting brain death. This score was routinely assessed and recorded in electronic medical records by well-trained physicians on duty of the emergency intensive care unit.”  (line 125-128)

The authors state that secondary outcomes include ARDS vs non-ARDS patient outcomes. This data is not presented.

Response>

We are sorry to make you confusing. We corrected typo and changed sentence as follow.

“Other ventilator parameters were compared between patients with good and poor neurologic outcome as secondary outcomes.”  (line 127-128)

Results:

Please add FiO2 values to Table 2. It would be helpful to have more complete ventilator data presented in the supplement, similar to the data presented in Table 2.

Response>

We added FiO2 values to Table 2 and more ventilator parameters on Table S1 (Additional file).

Table 2. Comparisons of ABGA and mechanical ventilator parameters on admission in the ARDS patients

Parameters

Total

(n = 119)

Poor outcome

(n = 96)

Good outcome

(n = 23)

p-value

pH

7.0 (6.9 – 7.2)

7.0 (6.8 – 7.2)

7.2 (7.0 – 7.3)

0.001

FiO2 (%)

100.0 (80.0 – 100.0)

100.0 (90.0 – 100.0)

100.0 (60.0 – 100.0)

0.028

PaO2 (mmHg)

86.0 (68.4 – 115.8)

86.0 (70.4 – 115.8)

77.1 (62.5 – 137.4)

0.287

PaCO2 (mmHg)

54.0 (40.0 – 73.0)

57.0 (43.5 – 76.0)

46.0 (32.6 – 60.5)

0.005

PEEP (cm H2O)

5.0 (4.0 – 8.0)

5.0 (4.0 – 8.0)

5.0 (4.0 – 7.0)

0.638

Inspiratory pressure (cm H2O)

15.0 (12.0 – 18.0)

16.0 (14.0 – 18.0)

12.0 (12.0 – 13.5)

< 0.001

Pplat (cm H2O)

21.0 (17.3 – 25.8)

21.0 (18.0 – 22.0)

17.0 (16.0 – 20.0)

< 0.001

Respiratory rate

20.0 (18.0 – 22.0)

20.0 (18.0 – 22.0)

18.0 (15.0 – 20.0)

0.013

Tidal volume (ml)

427.0 (367.0 – 497.0)

420.0 (360.0 – 489.5)

458.0 (395.5 – 545.0)

0.053

Compliance (ml/cm H2O)

29.7 (21.8 – 36.9)

27.5 (21.6 – 34.0)

38.6 (33.1 – 46.2)

< 0.001

Minute ventilation (L/min)

8.2 (7.6 – 10.4)

8.8 (7.7 – 10.3)

8.8 (6.5 – 10.7)

0.609

Data are presented as median with interquartile ranges.

Abbreviations: ABGA = arterial blood gas analysis; ARDS = acute respiratory distress syndrome; PEEP = positive end expiratory pressure; PIP = peak inspiratory pressure; Pplat = plateau pressure.

Discussion:

Paragraph one of the discussion could be improved by adding a brief summary of the predictive modeling results.

Response>

Thank you for your kindness. We accepted your opinion and changed paragraph one of the discussion section for summering our results.

“In this registry-based study, we found that lung compliance was associated with neurologic intact survival by time-dependent regression analysis. Maximum, minimum, and median compliance showed similar diagnostic performance for predicting neurologic outcomes and had the highest predictive value at day 1. Even though causality can be inferred only from randomized controlled trials, our finding suggested that the degree of the mechanical property of the respiratory system was an important marker for risk stratification.” (line 234-239)

The majority of the discussion is related to incidence of ARDS after OHCA which was not the objective of the paper. Consider revising. The discussion could be improved by a brief discussion related to the low NPV of the test.

Response>

Thank you for your kindness. We wanted to point out that the occurrence of ARDS after cardiac arrest was common and showed poor outcome before the insisting the importance of the compliance. We totally accept your opinion and rewrite discussion section about our results with low NPV of the test.

“When we set the cut-off value for each compliance, tests revealed high PPV with low NPV. A Although numerous studies have focused on discovering factors for poor neurologic outcome, it is essential to develop strategies for predicting good neurologic outcomes among OHCA survivors in order to appropriately tailor medical therapies for each patient [12,13]. (line 278-282)

The authors discuss change over 7 days (line 220) but do not show these results.

Response>

Sorry to make confusing. We showed trends of ventilator parameters in Table S1 (Additional file). As we mentioned, the differences and trends were not different after day 2. These results might be explained by numerous reasons. First, negative impact of occurring ARDS decides recovering neurologic recovery in initial phase and its degree become decreased. Second, measuring compliances are not possible when patients died and missing data of death patients could induce no differences between groups.

The authors state “These results imply that the recovery of lung and brain injury is determined in the initial phase…” While the pulmonary compliance may improve over the initial few days, the authors do not present any evidence related to brain injury recovery. This statement should be modified.

Response>

Thank you for your suggestion. Because of its retrospective design, it was impossible to clarify casualty between day-1 compliance and favorable outcome. However, many laboratory test results have proven that the interventions like TTM for preventing progression of hypoxic brain injury should be start before 24 hours because brain damage is aggravated after 24 hours. Our finding might be similar context with previous findings. We changed and added these evidences for making sentences more persuasive.

“Although casualty could not be confirmed, these results imply that the recovery of brain injury was determined in the initial phase, and proper management of ARDS would focus on the first 24 hours. This is consistent with evidence that brain damages started accelerating after 24 hours from global ischemia in laboratory animal and human tests [32,33].” (line 275-278)

The limitations section of the paper should reiterate the retrospective design and inability to differentiate association from causation.

Response>

We agreed with your opinion and added a sentence in limitation section.

“Moreover, definite casualty between compliance and neurologic outcome could not be confirmed.” (284-285)

Figure 1-

The text reference to figure 1 states: “For all seven compliances, day 1 showed significant large differences between the good and poor outcome groups that decreased over 7 days.” The figure does not show change over 7 days. This statement is confusing to me. Please label axes and legend.

Response>

We are sorry to make confusing. The sentence (“For all seven compliances, day 1 showed significant large differences between the good and poor outcome groups that decreased over 7 days.”) described the trend of compliance differences in Supplement table. Figure 1(now Figure 2, because of inserting Figure 1 for the flow chart) showed the differences of ROSC, day-1 maximum, median, and median compliances between good and poor outcome group. We corrected these mistakes. We also added label axes and legend.

“For all seven compliances, day 1 showed significant large differences between the good and poor outcome groups that decreased over 7 days.  Figure 2 showed the differences of ROSC, day-1 maximum, median, and minimum compliances between good and poor outcome group.” (line 205-208)

Minor comments:

Line 66: OHCA acronym incorrect.

Response> We corrected typo “OCHA” into “OHCA”.

“The OHCA registry consists of all adults (age ≥18 years) admitted to the emergency intensive care unit with non-traumatic CA.” (line 70)

Line 98: Authors state “missing data…means of each variable.” Please clarify whether this is patient means or population means. Also, the number of missing data points should be reported in the results.

Response>

It was the means of each patient, and we added words and each number of missing values to clarify the meaning.

“If there were missing ABGA or ventilator variable data (i.e. performed ABGA less than 3 times because patients visited ED just before the end of the day), the means of each variable were calculated by max and minimum values of each patient and filled before the statistical analysis (N = 9).” (line 111-114)

Line 163: missing the word “most”

Response> We inserted the word “most”.

“Among these variables, the minimum value was most significantly associated with the neurologic outcome (adjusted hazard ratios (HR) 1.08, 95% confidence interval (CI) 1.03 - 1.13; p < 0.001).” (line 194-196)

Add units to compliance measurements consistently throughout the manuscript.

Response> We corrected compliance measurements consistently throughout the article. (ml/cm H2O )

Round 2

Reviewer 1 Report

Regarding the paper ” Impact of lung compliance on neurological outcome 3 in patients with acute respiratory distress syndrome 4 following out-of-hospital cardiac arrest” submitted to JCM. I have the following comments.

The authors aim to draw general conclusions about compliance/oxygenation/carbon dioxide in a sub-set of OHCA patients. Of more interest would be to know more about why these OHCA patients developed ARDS. The authors should provide a table with the differences in baseline characteristics between those with ARDS and those without. It would be nice to know more about the progress of ARDS is these patients. I suspect this is a phenomenon of the first 48 hours. How many of these patients required prolonged mechanical ventilation specifically related to ARDS? In addition, it remains unclear which of the ARDS criteria were present in these patients. This is relevant. Which chest x-ray findings?My experience with OHCA patients is that many have infiltrates/low P/F ratios during the first 24 hours but this tends to resolve during the first 24 hours. Was there a difference in outcome between the ARDS/non ARDS patients? This is relevant. Most of the discussion about oxygen/carbon dioxide is problematic as these studies are conducted on a non-selected group of OHCA patients and the authors discuss a group selected based on oxygenation. This must be rectified. The authors cite several fairly old paper on oxygen after cardiac arrest. There have been two randomized trials on oxygen use after cardiac arrest. I suggest that the authors look up these and discuss them as they include better hints about whether high/low oxygen results in more severe brain injury (1,2) This study could be strengthened with more data on the cause of death in these patients. Why did they die? Neurological injury or shock? I am unsure about the relevance of Figure 2. ROC curves are relevant to biomarker studies when assessing prognostic accuracy. To me this is not the possible value of this study.

ICU-ROX Investigators and the Australian and New Zealand Intensive Care Society Clinical Trials Group, Mackle D, Bellomo R, Bailey M, Beasley R, Deane A, Eastwood G, Finfer S, Freebairn R, King V, Linke N, Litton E, McArthur C, McGuinness S, Panwar R, Young P. Conservative Oxygen Therapy during Mechanical Ventilation in the ICU. N Engl J Med. 2019 Oct 14. Jakkula P, Reinikainen M, Hästbacka J, Loisa P, Tiainen M, Pettilä V, Toppila J, Lähde M, Bäcklund M, Okkonen M, Bendel S, Birkelund T, Pulkkinen A, Heinonen J, Tikka T, Skrifvars MB; COMACARE study group. Targeting two different levels of both arterial carbon dioxide and arterial oxygen after cardiac arrest and resuscitation: a randomised pilot trial. Intensive Care Med. 2018 Dec;44(12):2112-2121.

Author Response

Regarding the paper ” Impact of lung compliance on neurological outcome 3 in patients with acute respiratory distress syndrome 4 following out-of-hospital cardiac arrest” submitted to JCM. I have the following comments.

The authors aim to draw general conclusions about compliance/oxygenation/carbon dioxide in a sub-set of OHCA patients. Of more interest would be to know more about why these OHCA patients developed ARDS. The authors should provide a table with the differences in baseline characteristics between those with ARDS and those without.

Response: According to your suggestion, we provided the baseline characteristics of the patients with and without ARDS. Older, hypertension, diabetes, lesser shockable rhythm, lesser emergent PCI, longer CPR duration, and higher SOFA score at admission were found in the ARDS group. We inserted this table in supplement material.

TABLE S2. Baseline characteristics of the whole study population

Characteristics

Total

(n = 246)

No ARDS

(n = 127)

ARDS

(n = 119)

P-value

Age

62.0 (48.8 – 74.0)

56.0 (43.0 – 71.0)

66.0 (55.0 – 78.0)

< 0.001

Male

159 (64.6)

80 (63.0)

79 (66.4)

0.596

Past history

Acute coronary syndrome

51 (20.7)

25 (19.7)

26 (21.8)

0.694

Arrhythmia

24 (9.8)

11 (8.7)

13 (10.9)

0.668

Congestive heart failure

15 (6.1)

6 (4.7)

9 (7.6)

0.429

Stroke

14 (5.7)

5 (3.9)

9 (7.6)

0.275

Hypertension

88 (35.8)

31 (24.4)

57 (47.9)

< 0.001

Diabetes mellitus

63 (25.6)

18 (14.2)

45 (37.8)

< 0.001

  Chronic pulmonary disease

19 (7.7)

7 (5.5)

12 (10.1)

0.233

Malignancy

27 (11.0)

15 (11.8)

12 (10.1)

0.689

Characteristics of Cardiac arrest

  Witnessed arrest

184 (74.8)

98 (77.2)

86 (72.3)

0.383

  Bystander CPR

169 (68.7)

94 (74.0)

75 (63.0)

0.074

Shockable rhythm

71 (36.4)

49 (49.0)

22 (23.2)

< 0.001

Cardiac cause

117 (47.6)

62 (48.8)

55 (46.2)

0.703

  Emergent PCI

28 (11.3)

19 (15.0)

9 (7.6)

< 0.049

Total CPR time (min)

24.0 (11.8 – 37.0)

21.0 (10. – 31.0)

29.0 (14.0 – 41.0)

0.005

SOFA at admission

11.0 (8.0 – 13.0)

8.0 (7.0 – 11.8)

12.0 (11.0 – 15.0)

< 0.001

Data are presented as n (%) or median with interquartile ranges. P<.05 are presented in bold.

Abbreviations: ARDS = acute respiratory distress syndrome; PCI = percutaneous coronary intervention; CPR = cardiopulmonary resuscitation; SOFA = Sequential Organ Failure Assessment.

It would be nice to know more about the progress of ARDS is these patients. I suspect this is a phenomenon of the first 48 hours. How many of these patients required prolonged mechanical ventilation specifically related to ARDS?

Response: We agreed with your concerns that the development of ARDS was resolved mostly within the first 48 hours. After excluding 27 patients (22.7%) who were died without improving ARDS within 48 hours, only 32 patients (26.9%) were resolved ARDS based on the Bering definition (PF ratio > 300 and resolved chest images) within 48 hours. And 60 (50.4%) were required prolonged mechanical ventilation due to ARDS (median 6 days [IQR 5 – 8.25]).

In addition, it remains unclear which of the ARDS criteria were present in these patients. This is relevant. Which chest x-ray findings?My experience with OHCA patients is that many have infiltrates/low P/F ratios during the first 24 hours but this tends to resolve during the first 24 hours.

Response: Based on the Berlin definition, we diagnosed ARDS with chest images, TTE, and PF ratio. All chest images (N = 119) were bilateral opacities which were recorded by the radiologist on duty. Among 119 patients of ARDS patients, only 32 patients (26.9%) were resolved during the first 24 hours.

Was there a difference in outcome between the ARDS/non ARDS patients? This is relevant.

Response: Thank you for your suggestions. We performed additional analysis for comparing clinical outcomes, such as 28-day mortality and neurologic outcome, and found that patients with ARDS showed a higher proportion of poor neurologic outcome (81.5% vs. 60.6%, p < 0.001) and higher all causes mortality (61.0% vs. 37.8%, p < 0.001). We added this finding in the Results section.

“Furthermore, patients with ARDS showed higher proportion of poor neurologic outcome (81.5% vs. 60.6%, p < 0.001) and higher all causes mortality (61.0% vs. 37.8%, p < 0.001).” (line 163-165)

TABLE. Comparisons of clinical outcomes between patients with or without ARDS

Outcomes

Total

(n = 246)

No ARDS

(n = 127)

ARDS

(n = 119)

P- value

Poor neurology outcome at day 28

174 (70.7)

77 (60.6)

97 (81.5)

< 0.001

All causes mortality at day 28

120 (49.0)

48 (37.8)

72 (61.0)

< 0.001

Data are presented as n (%) .

Abbreviations: ARDS = acute respiratory distress syndrome.

Most of the discussion about oxygen/carbon dioxide is problematic as these studies are conducted on a non-selected group of OHCA patients and the authors discuss a group selected based on oxygenation. This must be rectified.

Response: We agreed with your opinion that the role of O2 and CO2 for hypoxic brain injury was still controversial. According to your suggestion, we deleted the sentences and made it clear of different populations of the study groups.

Several animal studies indicate that hyperoxia early after the return of spontaneous circulation causes oxidative stress and harms post-ischemic neurons [25-27].

“These results could be explained by multifactorial reasons. First of all, lung compliance could reflect the severity of the ARDS. Numerous reports announced that patients with severe ARDS itself had higher mortality in critically ill patients, not only in patients with PCAS [12]. Moreover, in the landmark ARDS Network trial, limiting low Pplat which result in higher lung compliance improved outcome [26]. Second, severe ARDS aggravated the unbalance levels of oxygen and carbon dioxide, which could impair recovering neurocognitive function. However, the exact mechanisms of O2 and CO2 were not fully known. Although prior study showed post-resuscitation hyperoxia in the first 24 hour was associated with worse outcome [27], recent randomized trial revealed that different target level of PaCO2 (low-normal [33 – 35 mmHg] vs. high-normal [43 – 45 mmHg]) or PaO2 (normoxia [75 – 112 mmHg] vs. moderate hyperoxia [150 – 190 mmHg]) did not change the concentration of neuron-specific enolase at 48 hours after ROSC [28]. These mismatches might largely due to different inclusion of the study population or unmeasured confounders such as different prehospital settings [27,28].” (line 246-258)

The authors cite several fairly old paper on oxygen after cardiac arrest. There have been two randomized trials on oxygen use after cardiac arrest. I suggest that the authors look up these and discuss them as they include better hints about whether high/low oxygen results in more severe brain injury (1,2)

Response: We reviewed two randomized trials on oxygen use as you recommend. The study conducted by ICU-ROX investigators was not for the ROSC patients but for the general patients who were undergoing mechanical ventilation. On the other research, Jakkula et al. reported that there were no differences in the concentration of NSE at 48 hours after ROSC between the low and high levels of PaCO2 and PaO2. Although these results might imply that the gas imbalances could not affect the hypoxic brain injury, they only included witnessed OHCA of presumed cardiac cause with the initial shockable rhythm, which was a well-known population of favorable neurologic outcome. Moreover, the median serum level of NSE was around 20 ?g/l on both groups, and this concentration might not indicate the poor neurologic outcomes [1]. We deleted our old reference and rewrote it with a new reference in the discussion section.

“recent randomized trial revealed that different target level of PaCO2 (low-normal [33 – 35 mmHg] vs. high-normal [43 – 45 mmHg]) or PaO2 (normoxia [75 – 112 mmHg] vs. moderate hyperoxia [150 – 190 mmHg]) did not change the concentration of neuron-specific enolase at 48 hours after ROSC [28].” (line 253-256)

This study could be strengthened with more data on the cause of death in these patients. Why did they die? Neurological injury or shock?

Response: Thank you for your great suggestion. In our study population, 59 ARDS patients were in-hospital death. Of these, 37 patients (62.7%) were died due to ongoing multiorgan failure and refractory shock within 72 hours (i.e., they couldn’t finish TTM). Six patients (10.2%) had both severe hypoxic brain injury and progression of multiorgan failure. And, 16 patients (27.1%) stopped treatment because of severe hypoxic brain injury and expired. We added these in the result section.

Of 96 patients with poor outcomes, 59 ARDS patients were in-hospital death. Of these, 37 patients (62.7%) were died due to ongoing multiorgan failure and refractory shock within 72 hours (i.e., they couldn’t finish TTM). Six patients (10.2%) had both severe hypoxic brain injury and progression of multiorgan failure. And, 16 patients (27.1%) stopped treatment because of severe hypoxic brain injury and expired. (line 145-149)

I am unsure about the relevance of Figure 2. ROC curves are relevant to biomarker studies when assessing prognostic accuracy. To me this is not the possible value of this study.

Response: We agreed with your suggestion. We deleted the ROC curve (Figure 3).

Reference

Streiberger K.J., Leithner C, Wattenberg M., Tonner P.H., Hasslacher J., Joannidis M., Pellis T., Di Luca E., Fodlsch M., Krannich A., Ploner C.J., Storm C. Neuron-specific enolase predicts poor outcome after cardiac arrest and targeted temperature management: a multicenter study on 1,053 patients. Crit Care Med. 2017 56(7):1145-1151

Reviewer 2 Report

Introduction:

line 55: ... predict good neurologic outcomes among CA survivors.

Discussion:

You should discuss the influence of duration of CPR and shockable/non-shockable rhythm.

Conclusion:

The conclusion does not fit to the presented findings!

--> Taking into account shockable rhythm and duration of CPR lung compliance  may predict good neurologic outcome in CA survivors

Author Response

Introduction:

line 55: ... predict good neurologic outcomes among CA survivors.

Response: We corrected the sentence.

“Therefore, discovering accurate prognostic markers of ARDS severity is essential to facilitate risk stratification and apply novel therapeutic interventions to improve outcomes and predict neurologic outcomes among CA survivors.” (line 53-56)

Discussion:

You should discuss the influence of duration of CPR and shockable/non-shockable rhythm.

Response: CPR duration, shockable rhythm were well-known factors related with favorable outcomes. Our study population also showed shorter duration of CPR and higher proportion of shockable rhythm. Therefore, we included these confounders to regression model with compliance, and found compliance was associated with favorable outcome. We discussed this in the manuscript.

“In our result, favorable outcome group showed shorter duration of CPR and higher proportion of shockable rhythm which was well-known population of favorable neurologic outcome [3]. However, after modifying these effects by using regression analysis, we found that the higher compliances were associated with good outcome.” (line 258-261)

Conclusion:

The conclusion does not fit to the presented findings!

--> Taking into account shockable rhythm and duration of CPR lung compliance  may predict good neurologic outcome in CA survivors

Response: Thank you for your kind suggestion. We corrected the sentence in the conclusion as your offer.

“With shockable rhythm and short duration of CPR, lung compliance may be an early independent predictor of intact neurologic survival in ARDS patients following CA.” (line 306-307)
